# Usefulness of High-Sensitivity Troponin I in Risk Stratification and Final Disposition of Patients with Acute Heart Failure in the Emergency Department: Comparison between HFpEF vs. HFrEF

**DOI:** 10.3390/medicina59010007

**Published:** 2022-12-20

**Authors:** Luca Crisanti, Gabriele Valli, Elisa Cennamo, Alessandro Capolino, Paolo Fratini, Claudio Cesaro, Gloria Adducchio, Antonio De Magistris, Ferdinando Terlizzi, Maria Pia Ruggieri, Enrico Mirante, Claudio Savoriti, Kalyarat Sukruang, Valentina Valeriano, Francesco Rocco Pugliese, Francesco Travaglino, Salvatore Di Somma

**Affiliations:** 1Postgraduate School of Emergency Medicine, Sapienza University of Rome, 00189 Rome, Italy; 2Department of Emergency Medicine, San Giovanni Addolorata Hospital, 00184 Rome, Italy; 3Department of Emergency Medicine, University Campus Biomedico of Rome, 00128 Rome, Italy; 4Department of Emergency Medicine, Sant’Eugenio Hospital, 00144 Rome, Italy; 5Department of Family Medicine, Ramathibodi Hospital, Mahidol University, Bangkok 10400, Thailand; 6Department of Emergency Medicine, Sandro Pertini Hospital, 00157 Rome, Italy; 7Department of Medical Surgery Sciences and Translational Medicine, Faculty of Medicine and Psychology, Sapienza University of Rome, 00185 Rome, Italy; 8Global Research on Acute Conditions Team (Great Network), 00191 Roma, Italy

**Keywords:** acute heart failure, high-sensitivity troponin I, left ventricular ejection fraction, risk stratification, emergency department

## Abstract

*Background and Objectives:* In patients with acute heart failure (AHF), there is no definite evidence on the relationship between high-sensitivity cardiac troponin (hs-cTnI) and the left ventricular ejection fraction (LVEF) comparing the reduced and preserved EF conditions. *Materials and Methods*: Between January and April 2022, we retrospectively analyzed the data from 386 patients admitted to the emergency departments (ED) of five hospitals in Rome, Italy, for AHF. The criteria for inclusion were a final diagnosis of AHF; a cardiac ultrasound and hs-cTnI evaluations in the ED; and age > 18 yrs. We excluded patients with acute coronary syndrome (ACS). Based on echocardiography and hs-cTnI evaluations, the patients were grouped for (1) preserved (HFpEF) or (2) reduced LVEF (HFrEF) and a a) negative (within the normal range value) or b) positive (above the normal range value) of hs-cTnI, respectively. *Results*: There was a significant negative relationship between a positive test for hs-cTnI and LVEF. When compared to the group with a negative hs-cTnI test, the patients with a positive test, both from the HFpEF and HFrEF subgroups, were significantly more likely to have an adverse outcome, such as being admitted to the intensive care unit (ICU) or dying in the ED. Moreover, a reduced ejection fraction was linked with a final disposition to a higher level of care. *Conclusions*: In patients admitted to the ED for AHF without ACS, there is a negative relationship between hs-cTnI and a reduced LVEF, although a significant percentage of patients with a preserved LVEF also resulted to have high levels of hs-cTnI. In the absence of ACS, hs-cTnI seems to be a reliable biomarker of myocardial injury in AHF in the ED and should be considered as a risk stratification parameter for these subjects regardless of the left ventricular function. Further larger prospective studies are needed to confirm these preliminary data.

## 1. Introduction

Cardiovascular diseases are the main cause of death and complications worldwide [1]. Within these conditions, acute heart failure (AHF), which is a clinical syndrome characterized by a rapid onset of signs and symptoms that reflect an increase in intracardiac pressures or inadequate cardiac output, is a very frequent one [2]. According to the recent guidelines [2], AHF usually occurs as an acute decompensation of chronic heart failure with a rapid onset of symptoms, but it may also be the first manifestation of heart failure. The symptoms of AHF can be so severe to require urgent medical care, leading to ED admission and representing a frequent reason for hospitalization [3].

Each new episode of AHF increases the risk of mortality and morbidity, with a mortality rate to be around 5–10% within 30 days, independently of etiology or trigger events [4].

In order to start the treatment as soon as possible, the role of an emergency physician is crucial to improving the AHF patient outcome, because they are the front line operator for a fast and appropriate diagnosis and risk stratification [5]. Indeed, a significant relationship between the average waiting time before the visit and the number of deaths occurring in the ED has been described [6].

As a consequence, for AHF patients in the ED, identifying the appropriate final disposition and screening those patients that could be even discharged is also fundamental. Unfortunately, the guidelines reflect a lack of high-quality data [7] on how to achieve the goal in this effort, and although several clinical scores have been proposed, none of them seem to be able to predict hospital readmissions in the short term [8].

In AHF patients in the ED, a high-sensitivity cardiac troponin (hs-cTnI) evaluation is recommended to rule out ACS [2] and to identify the presence of myocardial injury [9]. When compared to conventional troponin I, hs-cTnI has also been found to contribute to the reduction in time needed to rule out/in ACS, leading to a reduction in the length and costs of hospitalization [10,11]. However, standard troponins levels could also be increased in no ACS-AHF and the increase in their concentration must be considered in the clinical setting of AHF as a predictor of mortality when linked with a reduced left ventricular ejection fraction [12]. Furthermore, other studies confirmed this association when a high-sensitivity troponin test was used [13,14].

Recently, the prognostic role of hs-cTnI in patients with chronic heart failure [15] and diastolic dysfunction [16] was described.

Therefore, in order to confirm the role of hs-cTnI as a marker of myocardial injury and risk stratification across the subgroups of left ventricular function, we decided to evaluate the relationship between hs-cTnI and the LVEF (both HFrEF or HFpEF) in patients with AHF and their final disposition from the ED.

## 2. Materials and Methods

### 2.1. Study Design and Population

This study was conducted according to the principles of the Helsinki Declaration. Data obtained from the clinical charts were recorded anonymously. Every patient included in this study gave consent to the analysis conducted in an anonymous manner.

For this study, we performed a retrospective analysis of the clinical data of five emergency departments (ED) in Rome, Italy. The hospitals involved were San Giovanni Addolorata hospital, Sant’Andrea hospital, Campus Bio-medico hospital, Sant’Eugenio hospital and Sandro Pertini hospital.

We included all admitted patients in any of the 5 ED from the 1st of January 2022 until the 30th of April 2022 with a final diagnosis of AHF. The diagnosis of AHF was made by the emergency physician according to the definition of European guidelines, confirmed by the measurement of natriuretic peptides and cardiac ultrasound [2]. The diagnosis was then confirmed by the investigators in each center after the review of clinical charts, before the enrollment. The echocardiography was performed by a consultant cardiologist during the visit in the ED. The same cardiologist was also involved in the clinical evaluation and in the exclusion of ACS, according to the fourth universal definition of myocardial infarction and European guidelines for n-STEMI [17,18]. Every hospital involved in the study had a predefined clinical algorithm for AHF evaluation and treatment based on the best clinical practice and current guidelines [2]. The concordance between the clinical algorithm for AHF management of the different hospitals was one of the criteria that was used for the inclusion of the hospital in the research.

In order to be included in the study, all the patients needed to have had at least an hs-cTnI determination at admission and a cardiac ultrasound evaluation in the ED. Our patients had to be at least eighteen years old. Based on medical history, physical exam, ECG and hs-cTnI level, we excluded patients with an ACS. If other diagnoses could explain signs and symptoms at presentation, those patients were not considered for the analysis. Finally, because there is no quality evidence regarding levels of hs-cTnI in patients with chronic kidney disease, they were also excluded from our cohort.

### 2.2. Clinical Information

Information regarding the admission and disposition date, triage code, final disposition, phenotype of heart failure, age and body weight were obtained from the medical charts of the ED. We also recorded all the data on vital signs at admission, such as blood pressure, heart rate, respiratory rate, SpO2, GCS and body temperature.

### 2.3. Biomarker Assay Measurements

All patients were tested in ED for hs-cTnI, brain natriuretic peptide (BNP), complete blood count (CBC), creatinine, sodium, potassium, calcium, BUN, PT and aPTT by the respective central labs of each hospital.

Concentrations of hs-cTnI were measured using the high-sensitivity immunoassay Abbott-Architect hs-cTnI (Abbott Diagnostics, Abbott Park, IL, USA). The upper reference limit was considered to be 34.2 ng/L for men and 15.6 ng/L for women.

### 2.4. Imaging

The cardiac ultrasound evaluation was performed by the cardiologist who was on duty in the ED. Quantitative measurements were taken according to the American Society of Echocardiography guidelines [19]. The degree of left ventricular dysfunction was assessed at three levels: preserved EF (HFpEF), when LVEF was over 50%; mild reduction EF (HFmrEF), when LVEF was between 40 and 50%; and reduced EF (HFrEF), if LVEF was less than 40%.

### 2.5. Outcome

As a measure of outcome, we used a composite endpoint including five possibilities: discharged, admitted to a low-intensity ward (i.e., cardiology, pneumology, internal medicine), admitted to a high-dependency unit (HDU), admitted to the intensive care unit (ICU), dead in the ED.

In order to simplify the analysis, patients admitted to ICU, HDU and those who died in the ED have been grouped together.

### 2.6. Statistical Analysis

The whole statistical examination has been performed using the statistical software StatPlus (StatPlus Pro v7©, AlaystSoft Inc., Walnut, CA, USA). Categorical variables were summarized in crosstab and expressed as % of the own group and analyzed using a χ^2^ test, and if the test gave a significant result, that was further analyzed with a z-test. Continuous variables were shown as median and interquartile range (IQR). The differences between medians were tested using the test U of Mann–Whitney. The Box-Plot graph was realized for the main results to visualize the values of distribution.

A two-sided *p* < 0.05 was considered statistically significant.

For the statistical analysis, we grouped together patients with an HFpEF (LVEF > 50%) and HFmrEF (LVEF between 41–49%), because we did not find any significant difference between the main variables (outcome; *p*-value 0.5, and hs-cTnI values; *p*-value 0.4). Furthermore, given the setting of ED, an ejection fraction of less than 40% was chosen as able to identify a compromised ventricular function, assuming that this value is less susceptible to variability between operators.

## 3. Results

### 3.1. Baseline Characteristics

We enrolled 386 patients admitted to the ED with a non ACS AHF between the 1st of January 2022 until the 30th of April 2022. Among those patients, 208 (54%) were males, average age 79.9 ± 10.7 sd years, with an uneven distribution across the three subgroups of the LVEF. The main triage code was 2 out of 5 (49.2%) and the second was code 1 (29.8%) [20]. The patients’ distribution across the five hospitals was normal, and there was no significant difference in the final disposition from the ED within the five centers. Based on the current guideline [2], 41.5% of our patients showed an HFrEF (LVEF < 40%), 27.7% an HFmrEF (LVEF between 41–49%) and the remaining 30.8% an HFpEF (LVEF > 50%), respectively.

The patient’s characteristics of our cohort, with the main phenotypes at admission and the final disposition from the ED, are shown in Table 1, while the clinical parameters of the patients according to the LVEF are described in Table 2.

### 3.2. hs-cTnI and LVEF

A total of 61% of the patients (236 patients) had a value of hs-cTnI higher than the upper reference limit. In the HFrEF patients (EF < 40%), the hs-cTnI values were significantly higher compared to the HFpEF (EF > 40%). We confirmed the negative relationship between the hs-cTnI and LVEF, as shown in Table 2. These results were valid both with the analysis of the absolute value of the hs-cTnI (Figure 1) and when using the dichotomous variable of a positive/negative test for the hs-cTnI.

### 3.3. Final Disposition

The patients with a positive test for hs-cTnI were significantly more likely to have an adverse outcome if compared to the group with a negative test (22% vs. 13%, *p* 0.02). Furthermore, all five patients who died in the ED had a positive test for hs-TnI (see Table 3 for more details). Regarding the ventricular function, one quarter of the patients with AHF and an LVEF below 40% had an adverse outcome (admitted to an HDU, ICU or died in the ED), while the same outcome was observed in 10% of the patients with a preserved LVEF, with a *p*-value between the three subgroups of 0.009. Figure 2 represents those patients with an adverse outcome, divided for the LVEF and the troponin test results.

## 4. Discussion

To the best of our knowledge, this is the first study that analyzes together the data of the hs-cTnI and the echocardiographic ventricular function in assessing the risk stratification in patients with AHF without ACS in the ED. The main findings of our retrospective study are as follows: (1) an hs-cTnI increase in patients with AHF and a more compromised ventricular function (Figure 1), (2) the high rate of admission to more intensive care of those patients with values of hs-cTnI above the upper limit of normality (Figure 2) and (3) the risk stratification value of hs-cTnI also for AHF subjects with a preserved LVEF.

The negative relationship between the LVEF and the hs-cTnI suggests that this biomarker might mirror the reduction in the ventricular function also when an ACS is excluded. Nevertheless, we still have to consider that the 59% of the patients with a preserved ejection fraction presented with levels of hs-cTnI higher than the upper reference limit. These findings could lead to a need for more intensive care for these subjects, as confirmed by the high percentage of patients admitted to the ICU and HDU and fewer patients discharged at home, and also for patients with HFpEF.

Furthermore, our data are partially in line with what was previously suggested by Peacock and coworkers [12], who observed values of conventional troponin significantly higher in patients with a lower LVEF. Compared to that study, where the standard cTn was used, we utilized the hs-cTnI. What we could also add to Dr. Peacock’s findings is that, utilizing a more sensitive test of TnI, there is also an important percentage of patients with a preserved LVEF who resulted to have a value of hs-TnI above the upper limit. From our result, it is not surprising that patients with a worse systolic function are more likely to be admitted to more intensive wards. From Figure 2, it seems that hs-cTnI has a higher prediction value for an adverse outcome if compared to the LVEF alone. In the introduction, we already mentioned that a prognostic value of hs-cTnI was described, but it is not sufficient to guide the clinicians to the final disposition. More detailed data together with prognostic information, obtained in a prospective study, will give us more certainties to rely on.

We should note that the severity of our patients’ symptoms at arrival is higher than those described in previous European studies. Almost 80% of our cohort was triaged as code 1 or 2 (respectively, “Red” and “Orange”, the two most serious codes that require to start treatment as soon as possible), while, for instance, Llorens et al. [4] published a cohort where most of the patients were triaged as level III (56.6%) or II (31.4%) of acuity. This might be a selection bias related to the retrospective design of the study. More severe patients are more likely to receive a cardiac ultrasound evaluation in the ED and, therefore, to meet the inclusion criteria for our study. The severity of our patients probably explains the important difference with other papers regarding the final disposition from our departments, because only 7.5% of the patients were discharged to home directly from the ED. We plan to avoid this bias with a prospective study where all the patients with AHF will receive an echocardiogram in the ED, therefore having a more representative cohort of the general population.

Regarding the natriuretic peptide, in our findings, there was a relationship between the BNP and LVEF, with the BNP values as high as the reduction in the LVEF and cardiac dysfunction. This relationship was desirable and confirms the role of the BNP as a marker of volume overload and cardiac wall stress (Table 2).

Nevertheless, Table 3 shows a significant increase in the BNP in patients with positive hs-cTnI values, confirming the role of natriuretic peptides to be considered in parallel to hs-cTnI as risk stratification biomarkers for AHF subjects, although a direct comparison between the BNP and hsTnI has been not performed in this study.

Finally, even if the echocardiography is always recommended in patients with AHF, usually it is deferred until the patient is stable enough to lay for the entire duration of the exam. It is possible that the implementation of a cardiac ultrasound in the ED, with a shorter and more focused exam, could help the physician to identify the appropriate setting of care.

### Limitations of the Study

This pilot study has several intrinsic limitations for the fact that it is a retrospective analysis with a heterogeneous population and a small sample. We could not make a proper risk stratification because we do not have follow-up data on our cohort after their disposition from the ED. Some patients with ACS may have been accidentally included in the study because the diagnosis was made at the discretion of the cardiologist. We also excluded a number of patients with non-ischemic heart failure because the echocardiogram was not performed in the ED.

Furthermore, in this retrospective study, we have not compared the risk stratification value of hsTnI with other biomarkers such as BNP, creatinine or sodium; therefore, we cannot exclude the possibility that other biomarkers that are routinely used in the assessment of patients with AHF could perform better or worse compared to hs-cTnI in the risk stratification of these subjects in the ED.

## 5. Conclusions

In patients with AHF in the ED, we confirmed the negative relationship between hs-cTnI and the LVEF. In the absence of ACS, hs-cTnI remains a reliable biomarker of myocardial injury in AHF, and it may mirror the left ventricular function. The final disposition from the ED correlates both with a positive test for hs-cTnI and a reduced LVEF, even though the troponin value probably has a primary role in the risk stratification because a high percentage of both LVEF subgroups, the preserved and reduced LVEF, showed a value of hs-cTnI above the upper limit. A prospective multicentric study with a longer follow-up will be needed in order to prove if a simultaneous cardiac ultrasound and hs-cTnI evaluation in the ED can be integrated for a more appropriate risk stratification in patients with AHF without ACS.

## Figures and Tables

**Figure 1 medicina-59-00007-f001:**
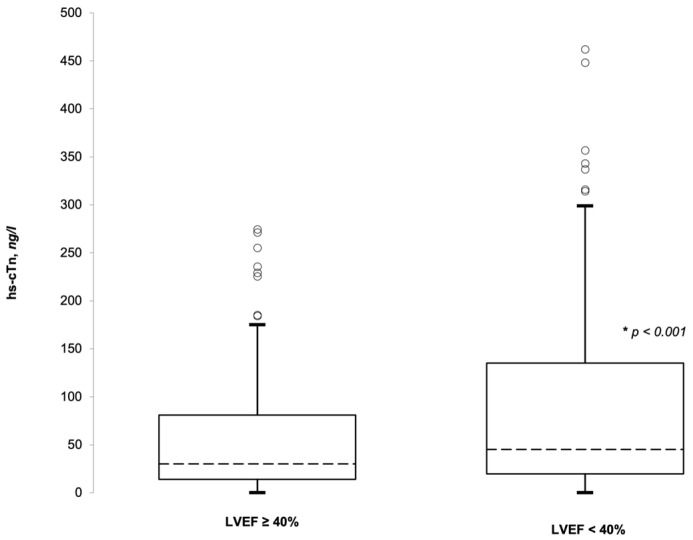
Absolute hs-cTnI values in patients with reduced and preserved ejection fraction. Median, IQR, 95% confidence range and extreme outlier data.

**Figure 2 medicina-59-00007-f002:**
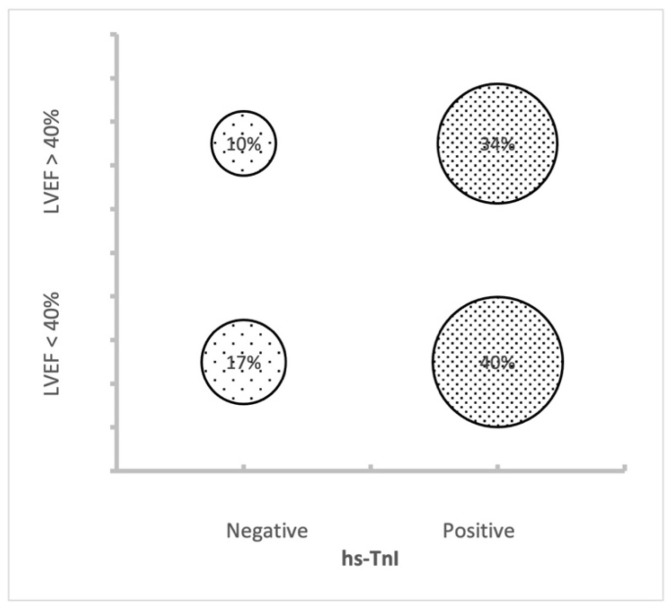
Distribution of patients with an adverse outcome divided for LVEF and hs-cTnI results.

**Table 1 medicina-59-00007-t001:** Patient’s characteristics (n.386).

Baseline Characteristics	n (%)	
Epidemiological data		
Age (years)	79.9	±10.7 SD
Male	208 (54)	
Triage		
Code 1 (red)	115 (29.8%)	
Code 2 (orange)	190 (49.2%)	
Code 3 (blue)	77 (19.9%)	
Code 4 (green)	4 (1.0%)	
Code 5 (white)	0 (0%)	χ^2^ < 0.001
Final Disposition		
Discharged	29 (7.5%)	
Low-intensity ward	286 (74.1%)	
High-dependency unit	44 (11.4%)	
Intensive care unit	22 (5.7%)	
Deceased	5 (1.3%)	χ^2^ < 0.001
Phenotype at presentation		
Acute decompensated heart failure	324 (83.9%)	
Acute pulmonary oedema	53 (13.7%)	
Isolated right ventricular failure	8 (2.1%)	
Cardiogenic shock	1 (0.3%)	χ^2^ < 0.001
Left ventricular ejection fraction		
EF < 40%	160 (41.5%)	
EF between 41 and 49	107 (27.7%)	
EF > 50%	119 (30.8%)	χ^2^ < 0.05
High sensitivity troponin I		
Positive test	236 (61.1%)	

EF, ejection fraction.

**Table 2 medicina-59-00007-t002:** Results according to LVEF.

	LVEF < 40%		LVEF > 40%		
n	160 (42%)		226 (59%)		
	Median	IQR	Median	IQR	*p*-Value
Age	81	73–85	82	76–88	<0.002
Systolic pressure	136	110–150	145	130–160	<0.01
Diastolic pressure	76	70–87	80	70–90	n.s.
Heart rate	85	75–99	85	75–98	n.s.
Respiratory rate	18	17–22	20	18–22	n.s.
Creatinine	1.19	0.92–1.7	1.11	0.88–1.55	n.s.
Hemoglobin	12.5	10.9–14.0	11.8	10.1–13.2	<0.01
La-	1,7	1.0–2.5	1.30	1.0–1.8	<0.05
BNP	2358	1060–8770	1200	587–2503	<0.001
hs-cTnI	42.3	18.1–116.1	29.3	13.9–73.0	<0.001
Deceased, n (%)	3 (2%)		2 (1%)		n.s.
Adverse outcome, n (%)	40 (25%)		31 (14%)		<0.01

**Table 3 medicina-59-00007-t003:** Results according to hs-cTnI test.

	hs-cTnI Negative		hs-cTnI Positive		
n	150 (39%)		236 (61%)		
	Median	IQR	Median	IQR	*p*-Value
Age	81	73–85	83	76–89	<0.001
Systolic pressure	140	120–150	140	125–160	n.s.
Diastolic pressure	80	70–86	80	70–90	n.s.
Heart rate	81	74–95	86	76–100	0.05
Respiratory rate	18	16–22	20	18–22	<0.05
Creatinine	1.08	0.88–1.38	1.23	0.95–1.71	<0.001
Hemoglobin	11.9	10.4–13.4	12.1	10.7–13.4	n.s.
La-	1.2	1.0–1.7	1.5	1.0–2.2	0.05
BNP	1192	595–2460	1797	752–4700	<0.001
hs-cTnI	12	9–18	66	38–143	<0.001
Deceased, n (%)	0 (0%)		5 (2%)		0.07
Adverse outcome, n (%)	19 (13%)		51 (22%)		0.02

## Data Availability

The data presented in this study are available on request from the corresponding author. The data are not publicly available due to patient confidentiality.

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
