# Peer review of "Usefulness of High-Sensitivity Troponin I in Risk Stratification and Final Disposition of Patients with Acute Heart Failure in the Emergency Department: Comparison between HFpEF vs. HFrEF"

_medicina, 2022, doi:10.3390/medicina59010007_

Round 1

Reviewer 1 Report

Interesting work addressing important issues from a practical point of view.  The main weakness is the retrospective nature of the analysis which generates limitations.

I would suggest adding information about the experience of the cardiologist who consulted the patients, including in limitations the addition of information regarding the different cardiologists who performed the measurements and the lack of early standardization of the cardiac evaluation.

I suggest considering expanding the literature.

Reviewer 2 Report

The authors analyzed the relationship between values of hs-troponin I and disposition in patients with acute heart failure in ER.

The overall composition of the presented work needs further refinement, and authors should self-check the manuscript before submission.  

Major concerns.

Introduction:

#1

There are numerous reports regarding the association between the elevation of troponin and worse outcomes in patients with acute heart failure. For example,

Pang, Peter S et al. “Use of High-Sensitivity Troponin T to Identify Patients With Acute Heart Failure at Lower Risk for Adverse Outcomes: An Exploratory Analysis From the RELAX-AHF Trial.” JACC. Heart failure vol. 4,7 (2016): 591-599. doi:10.1016/j.jchf.2016.02.009

Daniels, Lori B. “The enemy of good?: making the most of highly sensitive troponin assays.” Journal of the American College of Cardiology vol. 61,18 (2013): 1914-6. doi:10.1016/j.jacc.2013.01.065

Methods:

#2

The outcomes set by the authors are subject to biases. Say, if a ER doctor see higher value of troponin in patients with reduced ejection fraction, which ward he or she will chose for the patient to admit?

#3

If the authors tried to state the value of troponin level in risk stratification of patients with acute heart failure, they should show C-statistics. Does the value showed superiority compared to other existing examinations like BNP?

 #4

An ethical statment seems lacking in this study.

Discussion:

#5

What this study added to the field of risk stratification in patients with acute heart failure in the ER? They stated that “As consequence, for AHF patient in ED, identifying the appropriate setting of cure and to screen those patients that could be discharged and treated at home is also fundamental. Unfortunately, the guidelines reflect lack of high-quality data [7] on how to reach the goal in this effort and, although several clinical scores have been proposed, none of them seems to be able to predict the short term hospital readmissions [8].” Did their finding showed any superiority compared to those existing clinical scores?

Minor concerns:

#1

The English grammer needs extensive revision.

For example,

Page 2, lines 21 curecare

Page 8, first two paragraphs in the Discussion section are duplicated.

Page 11, References #1, what this yellow marking for?
